# Learning Linear Utility Functions From Pairwise Comparison Queries

## Abstract

There is increasingly widespread use of reward model learning from human preferences to align AI systems with human values, with applications including large language models, recommendation systems, and robotic control. Nevertheless, a fundamental understanding of our ability to successfully learn utility functions in this model remains limited. We initiate this line of work by studying learnability of linear utility functions from pairwise comparison queries. In particular, we consider two learning objectives. The first objective is to predict out-of-sample responses to pairwise comparisons, whereas the second is to approximately recover the true parameters of the utility function. We show that in the passive learning setting, linear utilities are efficiently learnable with respect to the first objective, both when query responses are uncorrupted by noise, and under Tsybakov noise when the distributions are sufficiently "nice". In contrast, we show that utility parameters are not learnable for a large set of data distributions without strong modeling assumptions, even when query responses are noise-free. Next, we proceed to analyze the learning problem in an active learning setting. In this case, we show that even the second objective is efficiently learnable, and present algorithms for both the noise-free and noisy query response settings. This qualitative learnability gap between passive and active learning from pairwise comparisons suggests that the tendency of conventional alignment practices to simply annotate a fixed set of queries may fail to yield effective reward model estimates, an issue that can be remedied through more deliberate query selection.

## 1 Introduction

Guiding machines in accordance with human values is a fundamental principle in technology, commonly referred to as "alignment" in the field of Artificial Intelligence (AI). A common approach for achieving alignment involves learning a utility function (or *reward model*) from a large collection of human responses to *pairwise comparison queries*, that is, queries about their preferences between pairs of options. A notable example of this is the collection of such data in training large language models (LLMs) to increase helpfulness and reduce harm (Ouyang et al., 2022; Bai et al., 2022) as part of a *reinforcement learning from human feedback (RLHF)* framework, in which learned reward models are used as reward functions in a reinforcement learning loop to fine-tune LLMs.

Despite the centrality of utility function estimation from pairwise comparison data across a broad array of domains (from personalized recommendation systems (Kalloori et al., 2018; Qomariyah, 2018) to kidney exchange platforms (Freedman et al., 2020) ), understanding of *which utility functions are learnable* remains limited. Theoretical consideration of this problem has been largely in the context of *random utility model (RUM)* learning based on complete preference information about a fixed set of candidates (outcomes) (Marschak, 1974; Becker et al., 1963; Negahban et al., 2018; Noothigattu et al., 2018). However, few have considered the issue of learning utility models over a vector space of outcomes, even when utility

functions are linear, where the observed preference data constitutes a negligible fraction of possible preference ranking data, and generalization beyond what is observed is key (a notable exception is Zhu et al. (2023), which we discuss below). In the real world, we often face an infinite choice space that the traditional framework fails to describe. As such settings constitute an important new horizon in social choice as it is integrated into modern AI, a deeper understanding of the issue is needed.

Our central goal is to initialize a comprehensive investigation of the problem of reward model learning that is central to value alignment, and particularly, the fundamental question *under what conditions is sample-efficient learning from pairwise comparison data possible.* We focus on linear utility models, but *allow arbitrary embeddings of the instance space* so that the class of utility models we consider is much more general (include the reward model used in LLM with neural network structure). Aside from a recent positive result by Zhu et al. (2023), who showed that parameters of a linear utility function can be efficiently estimated in the covariance norm when data follows the Bradley-Terry (BT) random utility model (Bradley & Terry, 1952), this question remains largely open. We extend their setting to the general random utility model of which BT, and many other common noisy utility models, are a special case. Moreover, we consider two natural measures of generalization error: 1) accuracy in *predicting* responses to unseen pairwise comparison queries, and 2) efficacy (which we capture as $\ell_p$ distances) in *estimating* parameters of the utility function.

We first investigate the "passive" learning setting in which pairs of inputs (outcomes) are generated according to an unknown distribution $\mathcal{P}$, and responses to these comparison queries may be corrupted by noise that comes from a given distribution $\mathcal{Q}$. When there is no noise in query responses, efficient learnability of learning to predict responses to unseen pairwise comparison queries follows directly from efficient learning of halfspaces (Blumer et al., 1989). However, the traditional PAC model allows so much freedom of the input distribution that the presence of any noise *with its c.d.f. continuous at* 0 can lead to an exponential lower bound on sample complexity. Nevertheless, we show that if we restrict the input distribution to be well-behaved, provided some mild conditions on the noise distribution, utilities are efficiently learnable in the PAC sense.

Turning next to the goal of estimating utility function parameters, we show that it is impossible to do so effectively with polynomial sample complexity *even when there is no query noise*. This is in contrast to the positive result by Zhu et al. (2023) which implies that we can efficiently learn linear utility functions when the smallest eigenvalue of the covariance matrix of the input distribution is bounded from below. A strong modeling assumption is thus necessary for learning in this sense.

While the "passive" setting is conventional in learning theory, the practice of reward model learning, particularly in the context of RLHF, allows one to carefully select the outcome pairs for which human preferences are queried (for example, we can curate both the prompts, and alternative responses to these, before sending the queries for annotation). We study this systematically by considering an *active learning* problem whereby we can select arbitrary pairs of inputs to query interactively in order to learn the parameters of a linear utility function. Our main results are that in the active learning model we can efficiently estimate parameters of the linear utility model, whether or not query responses are corrupted by noise. Our results thus suggest that conventional reward model learning approaches that construct the queries first, and subsequently have these annotated with pairwise comparison preferences, may yield potentially misleading estimates. In contrast, dynamically generating queries (such as prompts and pairs of responses) in a way that is designed to facilitate effective learning can lead to more reliable reward model estimation.

## 2 PRELIMINARIES

Our goal is to learn utility functions from pairwise comparison queries. To this end, we consider the hypothesis class $\mathcal{U}$ of weight-normalized monotone linear utility functions $u(x) = \vec{w}^T \phi(x)$ with (learnable) parameters $\vec{w}$, where $\phi(x) : \mathbb{R}^d \to \mathcal{X} \equiv [0,1]^m$ is a fixed and known feature function of candidate profiles

and $\mathbb{R}^d$ is the original space of candidates (defined by associated feature vectors $x$). More specifically, we consider utility functions with $\vec{w} \geq 0$ and $\|\vec{w}\|_1 = 1$, which has been the common assumption in RLHF Zhu et al. (2023). When the embedding is not needed, we just take $\phi(x) = x$. In many cases such as LLMs, $\phi$ is derived from a pretrained model with the last layer removed.

In the passive learning setting, we assume that we obtain a dataset $\mathcal{D} = \{(x_i, x_i', y_i)\}_{i=1}^n$ which contains labels $y_i \in \{0, 1\}$ associated with responses to pairwise comparison queries $(x_i, x_i')$ that are interpreted as preferences. In particular, $y_i = 1$ if $x_i'$ is preferred to $x_i$ (which we denote by $x_i' \succ x_i$), and $y_i = 0$ otherwise. We assume that input pairs $(x, x') \in \mathcal{D}$ are generated i.i.d. according to an unknown distribution $\mathcal{P}$ over $\mathbb{R}^d$. This, in turn, induces a distribution $\mathcal{P}_\phi$ over embedded pairs $(\phi(x), \phi(x'))$.

We assume that the label $y$ is generated according to the commonly used *random utility model (RUM)* (Noothigattu et al., 2020). Specifically, if $u(x)$ is the true utility function, query responses follow a random utility model $\tilde{u}(x) = u(x) + \tilde{\zeta}$, where $\tilde{\zeta}$ is an independent random variable distributed according to a fixed probability distribution $\tilde{\mathcal{Q}}$. Define $Z_{\tilde{u}}(x, x') \equiv \text{sign}(\tilde{u}(x') - \tilde{u}(x))$, where $\text{sign}(z) = 1$ if $z > 0$, $0$ if $z < 0$, and a random variable being $0$ or $1$ with equal probability $0.5$ if $z = 0$. Then $y = Z_{\tilde{u}}(x, x') \in \{0, 1\}$.

We further define the difference between a pair $(\phi(x), \phi(x'))$ as $\Delta_\phi(x, x') \equiv (\phi(x') - \phi(x)) \in [-1, 1]^m$. Since $\tilde{u}(x') - \tilde{u}(x) = \vec{w}^T \Delta_\phi(x, x') + \tilde{\zeta}(\phi(x')) - \tilde{\zeta}(\phi(x))$, it will be most useful to consider $\zeta \equiv \tilde{\zeta}(\phi(x')) - \tilde{\zeta}(\phi(x))$ for a pair of feature vectors $(\phi(x), \phi(x'))$, so that $Z_{\tilde{u}}(x, x') = \text{sign}(w^T \Delta_\phi(x, x') + \zeta)$. Let $\mathcal{Q}$ be the distribution over $\zeta$ induced by $\tilde{\mathcal{Q}}$, and let $F(\zeta)$ be its c.d.f.. Then the probability that $y = 1$ for a pair $(\phi(x), \phi(x'))$, that is, the probability that $x' \succ x$, is $\Pr(x' \succ x) = \Pr(\zeta \leq \vec{w}^T \Delta_\phi(x, x')) = F(\vec{w}^T \Delta_\phi(x, x'))$.

**Remark 1.** *Observe that for all $(x, x')$, $\Pr(x' \succ x) + \Pr(x \succ x') = 1$, i.e., $F(\vec{w}^T \Delta_\phi(x, x')) + F(-\vec{w}^T \Delta_\phi(x, x')) = 1$. Hence $F(x) + F(-x) = 1$ and $F(0) = 1/2$. These two conditions will be used for further analysis.*

Note that our random utility model is quite general. For example, the two most widely-used models are both special cases: the Bradley-Terry (BT) model (Bradley & Terry, 1952) corresponds to a logistic $F$, and the Thurstone-Mosteller (TM) model (Thurstone, 1927) corresponds to a Gaussian $F$. Additionally, we consider an important special case in which there is no noise, i.e., $\zeta = 0$. In this case, $\Pr(x' \succ x) \in \{0, 0.5, 1\}$ depending on whether $\vec{w}^T \Delta_\phi(x, x')$ is negative, zero, or positive.

Broadly speaking, the goal of learning utility functions from $\mathcal{D}$ is to effectively capture the true utility $u(x)$. In the setting where data provides only information about pairwise preferences, there are a number of reasonable learnability goals. Here, we consider two.

Our first goal is motivated by a general consideration that a common role of a utility function is to induce a ranking over alternatives, used in downstream tasks, such as reinforcement learning, recommendations, and so on. Thus, a corresponding aim of learning a utility function is to approximate its ranking over a finite subset of alternatives, so that the learned function is a useful proxy in downstream tasks. Here, we consider the simplest variant of this, which is to learn a linear function $\hat{u} : \mathcal{X} \to [0, 1]$ from $\mathcal{D}$ with the property that it yields the same outcomes from pairwise comparisons as $u(x)$. Formally, we capture this using the following error function:

$$e_1(\hat{u}, u) = \Pr_{(x, x') \sim \mathcal{P}} \left( Z_{\hat{u}}(x, x') \neq Z_u(x, x') \right). \tag{1}$$

We refer to this goal as minimizing the *error of predicted pairwise preferences*. Note that this error function has an important difference from conventional learning goals in similar settings: we wish to predict pairwise comparisons with respect to the true utility $u$, rather than the noise-perturbed utility $\tilde{u}$. This is a consequential difference, as it effectively constitutes a distributional shift when pairwise preference responses are noisy.

Our second goal is to accurately capture the weights of $u$. Formally, we define it as

$$e_2(\hat{u}, u) = \|\hat{\vec{w}} - \vec{w}^*\|_p^p \tag{2}$$

where $p \geq 1$ and $\vec{w}, \vec{w}^*$ are vectors denoting the weights of functions $\hat{u}, u$ respectively. Henceforth, we focus on Euclidean norm $\ell_2$. Our results generalize to any $p \geq 1$: counterexamples in the impossibility results still work, and the positive results hold by norm equivalence in $\mathbb{R}^m$. Moreover, focusing on $\ell_2$ norm facilitates a direct comparison to Zhu et al. (2023), who use a weighted 2-norm $\|x\|_\Sigma = \sqrt{x\Sigma x}$ that is most comparable with respected to the $\ell_2$. We refer to this model as the *utility estimation error*.

Our learnability discussion will be based on an adaptation of the *Probably Approximately Correct (PAC) Learning* framework (Valiant, 1984), with the goal of identifying a utility model $\hat{u}$ that has error at most $\varepsilon$ with probability at least $1 - \delta$ for $\varepsilon > 0$ and $\delta > 0$. We now formalize this as PAC learnability from pairwise comparisons (PAC-PC). Let $\mathbb{N}$ be the space of natural numbers.

**Definition 1** (PAC-PC learnability). *Given a noise distribution $\mathcal{Q}$, a utility class $\mathcal{U}$ is PAC learnable from pairwise comparisons (PAC-PC learnable) for error function $e(\hat{u}, u)$ if there is a learning algorithm $\mathcal{A}$ and a function $n_A : [0, 1]^2 \to \mathbb{N}$ such that for any input distribution $\mathcal{P}$, and $\forall \varepsilon, \delta \in [0, 1]$, whenever $n \geq n_A(\varepsilon, \delta)$, $\mathcal{A}(\{(x_i, x_i', y_i)\}_{i=1}^n)$ with $(x_i, x_i') \sim \mathcal{P}$ i.i.d. and $y_i = \text{sign}(u(x_i') - u(x_i) + \zeta_i)$ for $\zeta_i \sim \mathcal{Q}$ i.i.d. returns $\hat{u}$ such that $e(\hat{u}, u) \leq \varepsilon$ with probability at least $1 - \delta$. Moreover, if $n_A$ is polynomial in $1/\varepsilon, 1/\delta, \text{Dim}(\mathcal{U})$, where $\text{Dim}(\mathcal{U})$ denotes the VC-dimension of $\mathcal{U}$, then we say $\mathcal{U}$ is efficiently PAC-PC learnable.*

We stated this definition quite generally, but our focus here is on the class $\mathcal{U}$ of linear functions, in which case $\text{Dim}(\mathcal{U}) = m$.

## 3 PASSIVE LEARNING

We begin with the learning setting described as above. We refer to this as the *passive learning* setting to distinguish it from *active learning* that we consider later. For convenience, we define the distributions over $(\phi(x), \phi(x'))$ as $\mathcal{P}_\phi$ and that over $\Delta_\phi(x, x')$ as $\mathcal{P}_{\Delta_\phi}$, both induced by the input distribution $\mathcal{P}$ where pairs $(x, x')$ in the training data are generated i.i.d. from.

### 3.1 PREDICTING PAIRWISE PREFERENCES

We begin by considering the error function $e_1$, that is, where the goal is to predict outcomes of pairwise comparisons. In the noise-free setting, we can immediately obtain the following result as a direct corollary of learnability of halfspaces.

**Theorem 1.** *Suppose $\zeta = 0$. Excluding the input distribution where each pair satisfies that $\phi(x) - \phi(x') = 0$, the linear utility functions are efficiently PAC-PC learnable under the error function $e_1$.*

*Proof.* Since $y = \text{sign}(\vec{w}^{*T} \Delta_\phi(x, x'))$, PAC-PC learning under $e_1$ and $\mathcal{P}$ is equivalent to PAC learnability of halfspaces $\vec{w}^*$ under $\mathcal{P}_{\Delta_\phi}$. As shown by Blumer et al. (1989), halfspaces are learnable with sample complexity $\mathcal{O}(\frac{1}{\varepsilon}(m + \log(\frac{1}{\delta})))$. $\square$

We have excluded the special case where $\Delta_\phi(x, x') = 0$ holds across the dataset $\mathcal{D}$ because in the traditional learning literature, halfspaces are defined by a different sign function $\overline{\text{sign}}(x) = 1$ if $x \geq 0$ and $\overline{\text{sign}}(x) = 0$ if $x < 0$. For our setting, it would be impossible to learn to predict over entirely indistinguishable pairs whereas this problem is typically omitted for halfspace learning.

With the connection between learnability in the sense of $e_1$ and learning of halfspaces in mind, we now consider the case in which pairwise preference responses are corrupted by noise. We next show that for a

broad class of noise distributions, adding noise to pairwise preferences makes learning in the PAC-PC sense impossible.

For the next impossibility result, as well as the discussion that follows, it will be useful to define the chance of getting a misreported comparison label for $\Delta_\phi(x, x')$ as $\eta(\Delta_\phi(x, x'))$. Suppose the true label is $x \succ x'$, then $\vec{w}^{*T}\Delta_\phi(x, x') < 0$ and $\eta(\Delta_\phi(x, x')) = \Pr[x' \succ x] = F(\vec{w}^{*T}\Delta_\phi(x, x')) = F(-|\vec{w}^{*T}\Delta_\phi(x, x')|))$. Similarly, if the true label is $x' \succ x$, then $\vec{w}^{*T}\Delta_\phi(x, x') > 0$ and $\eta(\Delta_\phi(x, x')) = \Pr[x \succ x'] = F(-\vec{w}^{*T}\Delta_\phi(x, x')) = F(-|\vec{w}^{*T}\Delta_\phi(x, x')|))$. Hence, $\eta(\Delta_\phi(x, x'))$ is bounded above by $\frac{1}{2}$ as the c.d.f. $F$ is monotonically increasing and is equal to $\frac{1}{2}$ at 0, and $-|\vec{w}^{*T}\Delta_\phi(x, x')|$ is always non-positive.

**Theorem 2.** *If the preference noise distribution $\mathcal{Q}$ has a c.d.f. $F$ continuous at zero, $\mathcal{U}$ is not efficiently PAC-PC learnable under error function $e_1$.*

*Proof.* Suppose $F$ is continuous at 0: for any $t > 0$, there exists $s > 0$ such that for all $|v - 0| \leq s$, $|F(v) - F(0)| = |F(v) - \frac{1}{2}| \leq t$. Now for a fixed $\varepsilon < 1$, define $t_0 = 1/\exp(1/\varepsilon)$. Then there exists a corresponding $s_0$ such that for all $\vec{w}$ with the margin $|\vec{w}^T\Delta_\phi(x, x')| \leq s_0$, $|\eta(\Delta_\phi(x, x')) - \frac{1}{2}| \leq t_0$. Now consider the input distribution that puts all its weight on one pair $(x, x')$ that satisfies $|\vec{w}^{*T}\Delta_\phi(x, x')| \leq s_0$. Then we will witness the false label as a Bernoulli variable. With fixed probability $p \geq \frac{1}{2} - t_0$ we obtain the false label, and with probability $1 - p \leq \frac{1}{2} + t_0$ we obtain the true label. Recall that an optimal method to determine the true label is to query $n$ times and pick the majority label. (This event attains the TV distance between the given distribution and any distribution with the false label.) By Slud's inequality on Binomial distributions (Slud, 1977), the probability of witnessing the false labels for a majority of trials is bounded below by $\Pr[\sum_{i=1}^n \mathbb{I}\{\text{false label}\} > \frac{n}{2}] \geq \frac{1}{2}(1 - \sqrt{1 - \exp(-n(1 - 2p)^2/(1 - (1 - 2p)^2))})$. Suppose $n$ is in $o(1/(1 - 2p)^2)$, then $\Pr[\sum_{i=1}^n \mathbb{I}\{\text{false label} > \frac{n}{2}\}]$ will be $o(1)$, i.e., below any positive constant, since for any fixed $p \in (0, \frac{1}{2})$, $\frac{1}{2}(1 - \sqrt{1 - \exp(-1/(1 - (1 - 2p)^2))})$ is a positive constant. Therefore, in order to establish confidence with arbitrary magnitude, the sample complexity needs to be $\Omega(1/(1 - 2p)^2) = \Omega(1/(p - 1/2)^2)$. As $0 < 1/2 - p \leq t_0$, $\frac{1}{(p - 1/2)^2} \geq \frac{1}{t_0^2}$. Consequently, the sample complexity is in $\Omega(\frac{1}{t_0^2}) = \Omega(\exp(\frac{1}{\varepsilon})^2)$, exponential in $\frac{1}{\varepsilon}$. $\square$

Notably, all common models of noisy preference responses, such as Bradley-Terry and Thurstone-Mosteller, entail continuity of $F$ at 0, so this result yields impossibility of learning linear utilities for these standard RUM settings. The consequence of Theorem 2 is that to achieve any general positive results we must constrain both the distribution over inputs, and the noise to some degree.

We now show that we can leverage Tsybakov noise and a *"well-behavedness"* condition on distributions to attain sufficient conditions for efficient learnability.

**Definition 2** (Tsybakov Noise Condition). *(Tsybakov, 2004) Let $C$ be a concept class of Boolean-valued functions over $X = \mathbb{R}^d$, $F$ be a family of distributions on $X$, $0 < \varepsilon < 1$ be the error parameter, and $0 \leq \alpha < 1$, $A > 0$ be parameters of the noise model. Let $f$ be an unknown target function in $C$. A Tsybakov example oracle, $EX^{Tsyb}(f, F)$, works as follows: Each time $EX^{Tsyb}(f, F)$ is invoked, it returns a labeled example $(x, y)$, such that: (a) $x \sim \mathcal{D}_x$, where $\mathcal{D}_x$ is a fixed distribution in $F$, and (b) $y = f(x)$ with probability $1 - \eta(x)$ and $y = 1 - f(x)$ with probability $\eta(x)$. Here $\eta(x)$ is an unknown function that satisfies the Tsybakov noise condition with parameters $(\alpha, A)$. That is, for any $0 < t \leq \frac{1}{2}$, $\eta(x)$ satisfies the condition $\Pr_{x \sim \mathcal{D}_x}[\eta(x) \geq \frac{1}{2} - t] \leq At^{\frac{\alpha}{1-\alpha}}$.*

**Definition 3** (Well-Behaved Distributions). *(Diakonikolas et al., 2021) For $L, R, U > 0$, $k \in \mathbb{Z}_+$, and $\beta \geq 1$, a distribution $\mathcal{D}_x$ on $\mathbb{R}^d$ is called $(k, L, R, U, \beta)$-well-behaved if (i) for any $t > 0$ and unit vector $\vec{w} \in \mathbb{R}^d$, we have that $\Pr_{x \sim \mathcal{D}_x}[|\langle w, x \rangle| \geq t] \leq \exp(1 - t/\beta)$ (subexponential concentration) and for any projection $(\mathcal{D}_x)_V$ of $\mathcal{D}_x$ on a $k$-dimensional subspace $V$ of $\mathbb{R}^d$, the corresponding p.d.f. $\gamma_V$ on $V$ satisfies*

*the following properties: (ii) $\gamma_V(x) \geq L$, for all $x \in V$ with $||x||_2 \leq R$ (*anti-anti-concentration*), and (iii) $\gamma_V(x) \leq U$ for all $x \in V$ (*anti-concentration*).*

Although the definition may appear challenging to understand, we believe Tsybakov noise is ideal for modeling human judgments. This family of noise processes is analytically tractable and allows noise rates to vary based on the distance from the decision boundary in any latent space. It imposes only a bound on the density of points with a given noise rate, rather than on their actual geometry, and approaches pure noise only at the boundary. As the only standard model with these properties, it is particularly suited for situations where there is no strong preference between alternatives that are essentially equivalent in terms of utilities. Moreover, it has also been adopted by a different branch of learning from pairwise comparison such as Xu et al. (2017).

And the intuition behind this seemingly convoluted definition strikes a balance between avoiding excessive concentration (as having all the mass on one extremely similar pair would render the data uninformative) and insufficient concentration (ensuring that we can still derive meaningful insights from the observations). Indeed, Diakonikolas et al. (2021) have demonstrated how to learn halfspaces efficiently in the presence of Tsybakov noise, under the following *well-behavedness* condition:

**Theorem 3** (Learning Tsybakov Halfspaces under Well-Behaved Distributions)**.** *(Diakonikolas et al., 2021, Theorem 5.1) Let $\mathcal{D}_x$ be a $(3, L, R, U, \beta)$-well-behaved isotropic distribution on $\mathbb{R}^d \times \{\pm 1\}$ that satisfies the $(\alpha, A)$-Tsybakov noise condition with respect to an unknown halfspace $f(x) = \text{sign}(\langle w, x \rangle)$. There exists an algorithm that draws $N = \beta^4 (\frac{dUA}{RL\varepsilon})^{O(1/\alpha)} log(1/\delta)$ samples from $\mathcal{D}_x$, runs in $poly(N, d)$ time, and computes a vector $\hat{w}$ such that, with probability $1 - \delta$, we have $\Pr_{x \sim \mathcal{D}_x}[h(x) \neq f(x)] \leq \varepsilon$.*

By leveraging the condition on the input distribution as well as a mild assumption on the noise distribution, we can establish the conditions of the Tsybakov model hold. This leads to the following positive result, where poly($\cdot$) means polynomial in the argument:

**Theorem 4.** *Suppose that $\mathcal{P}_\phi$ is a $(3, L, R, U, \beta)$-well-behaved isotropic distribution, and the noise c.d.f. $F^{-1}(\zeta) \leq \text{poly}(\zeta)$ on $(0, \frac{1}{2}]$. Then the linear utility class $\mathcal{U}$ is efficiently PAC-PC learnable.*

Due to the length of the proof, it is deferred to Appendix A, accompanied by our demonstration that the standard models of noise in RUM settings—Bradley-Terry (Bradley & Terry, 1952) and Thurstone-Mosteller (Thurstone, 1927)—both satisfy the condition on the noise distribution in Theorem 4.

## 3.2 ESTIMATING UTILITY PARAMETERS

Next, we tackle the more challenging learning goal represented by the error function $e_2$, that is, where our goal is to learn to effectively estimate the parameters $\vec{w}$ of the true linear utility model in the $\ell_p$ sense (focusing on $\ell_2$ here for clarity of exposition).

We begin with the known positive result. Specifically, Zhu et al. (2023) showed that maximum likelihood estimation (MLE) with the common BT noise model achieves efficient utility estimation in the following sense. With probability at least $1 - \delta$, the MLE parameter estimator $\vec{w}$ for the loss $\ell_{\mathcal{D}} = -\frac{1}{n} \sum_{i=1}^{n} \log \left( 1(y^i = 1) \cdot \Pr(x' \succ x) + 1(y^i = 0) \cdot \Pr(x \succ x') \right)$ from $n$ samples has a bounded error measured in a seminorm with respect to $\Sigma = \frac{1}{n} \sum_{i=1}^{n} (\phi(x_i) - \phi(x_i')_i)(\phi(x_i) - \phi(x_i'))^T$. With $C$ being constant,

$$||\hat{\vec{w}} - \vec{w}^*||_\Sigma = \sqrt{(\hat{\vec{w}} - \vec{w}^*)^T \Sigma (\hat{\vec{w}} - \vec{w}^*)} \leq C \cdot \sqrt{\frac{m + \log(1/\delta)}{n}}. \tag{3}$$

We are able to extend their positive result to a larger set of RUMs. The proof is deferred to the Appendix B due to the similarity of the approaches.

**Theorem 5.** *If there exists $\gamma > 0$ such that the noise c.d.f. $F(z)$ satisfies $F'(z)^2 - F''(z) \cdot F(z) \geq \gamma$ for all $z$, then with probability $1 - \delta$, the MLE estimator $\hat{\vec{w}}$ satisfies inequality (3).*

Consequently, as long as $\Sigma$'s smallest eigenvalue is bounded from below (which excludes the case where $\phi(x), \phi(x')$ are consistently close, as $\Sigma$ tends to 0 the bound becomes vacuous), sample complexity with respect to $e_2$ will also be quadratic. Formally, suppose that the smallest eigenvalue $\lambda_{min} \geq C_2 > 0$. Then,

$$||\hat{\vec{w}} - \vec{w}^*||_2 = \sqrt{\frac{C_2}{C_2}(\hat{\vec{w}} - \vec{w}^*)^T(\hat{\vec{w}} - \vec{w}^*)}$$

$$\leq \sqrt{\frac{1}{C_2}(\hat{\vec{w}} - \vec{w}^*)^T \Sigma (\hat{\vec{w}} - \vec{w}^*)}$$

$$\leq \overline{C} \cdot \sqrt{\frac{m + \log(1/\delta)}{n}}$$

for a constant $\overline{C}$.

However, the above condition $F'(z)^2 - F''(z) \cdot F(z) \geq \gamma$ is both hard to interpret and hard to satisfy (Thurstone-Mosteller fails this condition for instance). Now we show that *without any structural noise assumption,* learning to estimate linear utility parameters in the $\ell_2$ norm is impossible, even when we have no noise at all:

**Theorem 6.** *If there is no noise, the class of linear utility functions $\mathcal{U}$ is not PAC-PC learnable with respect to $e_2$, i.e. there is a probability distribution $\mathcal{P}$ over $(x, x')$, that no learning algorithm can achieve $e_2 \leq \varepsilon$ with probability at least $1 - \delta$, for any $0 < \varepsilon, \delta < 1$ under a finite set of samples.*

*Proof.* Consider an oracle that only provides pairs $(x, x')$ satisfying $\phi(x')^i > \phi(x)^i$ on each dimension $i$. Then since all $\vec{w} \geq 0$, $\vec{w}^{*T} \Delta_\phi(x, x') \geq 0$ for every input pair $(x, x')$; in other words, the labels are uninformative because we can also infer the sign of $\vec{w}^{*T} \Delta_\phi(x, x')$ being positive and any algorithm produces the same distribution on guesses $\hat{\vec{w}}$ for all $\vec{w}$. For $\delta < \frac{1}{2}$, we can consider $\vec{w}, \vec{w}' \in \mathcal{U}$ with distance equal to $\sqrt{2}$. This is achievable because $\mathcal{U}$ contains segments of length $\sqrt{2}$ defined by $\{w^i + w^j = 1, 0 \leq w^i, w^j \leq 1\}$ for any distinct pair of $i, j \in \{1, \ldots, m\}$ and we know $m \geq 2$ for otherwise there is no need to learn. But the algorithm cannot output $\hat{\vec{w}}$ with $e_2 < (\frac{\sqrt{2}}{2})^2 = \frac{1}{2}$ for both $\vec{w}$ and $\vec{w}'$ simultaneously, and hence its output must obtain $e_2 \geq \frac{1}{2}$ with probability $\geq \frac{1}{2}$ for one of these. $\qquad\square$

The reason behind this contrast between the positive and negative results lies precisely in the assumption of the noise model. Unlike a common misconception that noise always makes learning more challenging, a highly structured model assumption like BT actually provides more information to each query — from a pair where $\Delta_\phi(x, x') > 0$, if we know $F$, and we can estimate the probability of its label being $x' \succ x$ as $\Pr(x' \succ x)$, since $\Pr(x' \succ x) = F(\vec{w}^{*T} \Delta_\phi(x, x'))$, we obtain information about $\vec{w}^*$ immediately.

To illustrate a necessary and sufficient results for passive learning in the noise-free or noise-agnostic setting, let us switch to a geometric perspective. It is helpful to recall the learning theoretic concept of the *version space*. A version space consists of all "surviving" hypotheses that are consistent with the labeled examples observed so far. In our case, our version space denoted as $\mathcal{W}$ will be initialised the same as $\mathcal{U}$. Now as each datapoint comes along, the label $y$ gives us an (inaccurate) sign of $\vec{w}^{*T} \Delta_\phi(x, x')$, indicating whether the vector $\vec{w}^*$ that we are searching for is above, on, or below the hyperplane defined by its normal vector $\Delta_\phi(x)$. Then the version space can update accordingly. For example, when $m = 3$, the initial version space (hypothesis class) is depicted as the triangle $\{w^1 + w^2 + w^3 - 1 = 0 \mid 0 \leq w^1, w^2, w^3 \leq 1\}$ in Figure 1, with the true parameter labeled as $\vec{w}^*$. Then suppose we receive a noiseless datapoint labeled with $y < 0$,

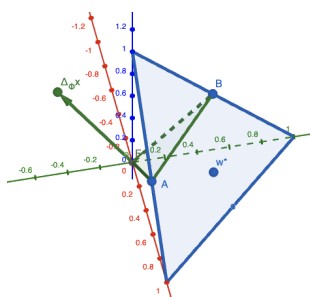

Figure 1: From the label we know $\vec{w}^*$ is below the hyperplane of $\Delta_\phi(x, x')$.

we can infer $\vec{w}^*$ is below the segment $AB$ drawn as in Figure 1 since that is the space $\Delta_\phi(x, x')$ is normal to.

Thus, we require that the incoming datapoints shrink the version space into a $\varepsilon$-radius ball centred at $\vec{w}^*$. The difficulty with respect to $e_2$ is exactly that we don't have any control over the incoming datapoints. Hence we will see in the following section that active learning is much more advantageous.

## 4 ACTIVE LEARNING

Thus far, we have noted that passive learning is typically quite challenging, especially when responses to pairwise comparison queries are noisy and there is no precise noise model assumption. This has led to more recent investigations into active learning in RLHF (Liu et al., 2024; Guo et al., 2024; Das et al., 2024; Muldrew et al., 2024). While these models are motivated by the practice of large language models (LLMs), they do not generalize to all random utility models (RUMs). In this section, we demonstrate that the active learning approach in this setting differs qualitatively from passive learning, yielding stronger and more efficient learnability results.

Our investigation falls under the umbrella of an interactive approach, which is considered a more efficient framework than pool and stream-based sampling techniques (Ling & Du, 2008; Alabdulmohsin et al., 2015; Wang & Singh, 2016; Chen et al., 2017). A common argument against query synthesis methods is that the artificial queries we thereby generate are uninterpretable. However, there is evidence that generating highly informative artificial training instances for tasks like text classification is feasible (Schumann & Rehbein, 2019; Piedboeuf & Langlais, 2022). Although more empirical tricks may need to be developed to apply active learning approaches practically, these can in general save considerable annotation labor. We structure our investigation in the same way as the passive learning case, considering first the problem of learning to predict responses to pairwise preference queries (the $e_1$ error model), and subsequently dealing with the more challenging problem of estimating utility parameters.

### 4.1 PREDICTING PAIRWISE PREFERENCES

Just as in the passing learning setting, the positive results for active learning of linear utilities from pairwise comparisons in the $e_1$ sense follow directly from known results for learning halfspaces. In the noise-free case, Alabdulmohsin et al. (2015) and Chen et al. (2017) provide efficient learning algorithms through query synthesis. Zhang & Li (2021), in turn, address the setting with Tsybakov noise. Consequently, we focus on the more challenging problem of estimating utility parameters in the active learning setting. We tackle this problem next.

## 4.2 ESTIMATING UTILITY PARAMETERS

Continuing from our discussion in Section 3.2, the goal in active learning is to ensure that in each step we query the most informative comparison pair inside the current version space in order to reduce the size of the version space as fast as possible. We first establish an efficient algorithm for this in the noise-free setting, and subsequently in the case with query noise. Here, a key challenge is the ability to invert the embedding $\phi(x)$. Since our focus is on *query* or *sample* complexity, our results hold whether or not computing an inverse of $\phi$ (or, equivalently, a zero of $\phi(x) - v$ for a given $v$) is efficient (note that we do not need uniqueness); however, this does, of course, impact *computational* complexity of the algorithms. In special cases, such as if $\phi(x) = x$, or if $\phi(x)$ is affine (where an inverse can be computed using linear programming), computational complexity is polynomial as well, and more generally, we can in practice use gradient-based methods (such as Newton's method) to approximately find a zero of $\phi(x) - v$.

Specifically, we aim to enclose the true weight $w^*$ within a $m - 1$ dimensional hypercube with side length $\frac{2\varepsilon}{\sqrt{m-1}}$, such that the longest possible distance between the hypercube's center and any point in the cube is the half diagonal $\frac{\sqrt{m-1}}{2} \cdot \frac{2\varepsilon}{\sqrt{m-1}} = \varepsilon$. Then, by picking the center of the cube, we can ensure the error is bounded as $e_2 \leq \varepsilon$. Thus, our algorithm runs like a binary search on each of the $m - 1$ dimensions. In order to shrink the length of the searching space in each dimension, we recursively select the query $(x, x')$ with difference $\Delta_\phi(x, x')$ being (approximately) the normal vector of the hyperplane halving the original space along that dimension. If there is no noise, our algorithm can bound the estimation error with probability 1. If noise exists, we make each query more times, and take the majority vote as the true comparison label.

Pseudocodes of the algorithms are deferred to Appendix C and D, where we also prove their sample efficiency.

**Theorem 7.** *Suppose $\zeta = 0$ and we can approximate the inverse of $\phi$ with $\tilde{\phi}^{-1}$ up to arbitrary precision, i.e. $||\phi(\tilde{\phi}^{-1}(x)) - x||_\infty < c$ for any constant $c$. Then for any $\varepsilon$, there is an active learning algorithm that returns a linear hypothesis $\hat{u}$ with $e_2(\hat{u}, u) \leq \varepsilon$ after asking the oracle $\mathcal{O}(m \log(\frac{\sqrt{m}}{\varepsilon}))$ queries.*

**Theorem 8.** *For a fixed $\mathcal{Q}$ with corresponding c.d.f. $F$, suppose we can approximate the inverse of $\phi$ with $\tilde{\phi}^{-1}$ up to arbitrary precision, i.e. $||\phi(\tilde{\phi}^{-1}(x)) - x||_\infty < c$ for any constant $c$, then there exists an active learning algorithm that outputs $\hat{u}$ with $e_2(\hat{u}, u) \leq \varepsilon$ with probability at least $1 - \delta$ after $\mathcal{O}(\frac{1}{(p_0 - 1/2)^2} \log(\frac{1}{\delta}))$ queries where $p_0 = F(\frac{\varepsilon}{\sqrt{m-1}}) > 1/2$.*

The efficiency result in Theorem 8 thus depends on the noise c.d.f. $F(x)$. In the case of the logistic noise model based on the Bradley-Terry distribution, this leads to a better sample complexity than the passive learning model of Zhu et al. (inequality (3)). The derivation is provided in Appendix D, along with a simple numerical illustration(2) showcasing the sample complexity gap between passive and active learning approaches.

## 5 RELATED WORK

**Learning Utility Models from Pairwise Comparisons:** The fundamental framework that we build on is random utility model learning, in which utility information is provided indirectly through ranking (e.g., pairwise) comparisons (Marschak, 1974; Bradley & Terry, 1952; Xia, 2019). Previously, "learning to rank" (Liu et al., 2009) has been extensively explored for the application of information retrieval. These algorithms can be grouped into three approaches: *pointwise*, *pairwise*, *listwise*. Our first learning goal is most similar to their *pairwise* learning goal, as it involves producing pairwise comparisons of candidates, and our second learning goal aligns with their *pointwise* learning goal, as it involves estimating the utility (or relevance degree in the information retrieval's context) of a single point. However, their learning inputs are usually not

pairwise comparisons. On the other hand, there is a branch of work on learning *with* pairwise comparisons (Xu et al., 2017; Zeng & Shen, 2022). Unlike our setting, they typically assume comparison data as additional information rather than the single data source. The limited work in the same setting all focuses on a finite set of alternatives (Bong & Rinaldo, 2022; Li & Zhang, 2023; Shah et al., 2015; Negahban et al., 2018; Jamieson & Nowak, 2011). Even though we can recast our model into the one studied in these paper as sparse "one-hot" features, they do not face the generalization challenge we address.

**Reinforcement Learning from Human Feedback** Unlike the traditional social choice community's set-up, the recent practice of RLHF concerns itself with the generalization problem. However, their attention has largely been limited to the Bradley-Terry model, which is just a special case within our broader consideration. Indeed, in the passive learning set-up, the specific model has been considered by Zhu et al. (2023). As for active learning, few empirical studies have demonstrated the advantage of online human feedback (Muldrew et al., 2024; Guo et al., 2024). However, our work remains the first theoretical treatment of the subject. In particular, we unify the framework of computational social choice with the RLHF practice.

**PAC Learning of Halfspaces With Noise:** Common models of noise in random utility models have the property that the closer the pair's difference is to the halfspace defined by the linear utility function, the higher the chance that this comparison is flipped. This kind of noise has been previously studied as boundary-consistent noise in (Du & Cai, 2015; Zhang et al., 2021; Zhang & Li, 2021). The possibility of an arbitrarily small difference between noise and $\frac{1}{2}$ makes learning very challenging, even for learning halfspaces (Balcan & Haghtalab, 2020). Consequently, proposals for tractable learning often rely on assumptions regarding distributions of noise. The spectrum of such noise models starts from the easiest version called *random classification noise (RCN)* (Angluin & Laird, 1988) to the other end of malicious noise (Kearns & Li, 1988) and agnostic learning (Kearns et al., 1992; Klivans & Kothari, 2014; Daniely, 2016; Diakonikolas et al., 2020). Additionally, active learning has been proposed as a solution to reduce the number of samples the learner needs to query before approximately learning the concept class with high probability. While active learning requires exponentially fewer labeled samples than PAC-learning for simple classes such as thresholds in one dimension, it fails in general to provide asymptotic improvement for broad classes such as halfspaces (Dasgupta, 2005).

**Robust Parameter Estimation:** Learning a linear classifier has also been a classical problem beyond learning theory, with *empirical risk minimisation* the most popular paradigm (Vapnik, 1991), with a range of tools including Bayes classifiers, perceptron learning, and support vector machines (SVM). Most directly related is the connection between robustness (to input noise) and regularization in SVM (Xu et al., 2009). However, this robustness is with respect to *input* noise, whereas our consideration is noise in pairwise comparison responses (outputs).

# 6 CONCLUSION AND LIMITATIONS

We presented a theoretical investigation into the learnability of linear utility functions from pairwise comparison queries. Our results consider both the passive and active learning problems, as well as two objectives: the first involving *prediction* and the second concerned with *estimation*. Overall, we find that estimation is generally more challenging than prediction, and active learning enables qualitatively better sample complexity in this case. There are several directions to extend our work. For example, one could aim to generalize our results to list-wise comparisons as described in Zhao & Xia (2019). Specific to active learning, especially related to LLM and RLHF, we acknowledge the challenges of query synthesis and the issue of inverting the embedding $\phi(x)$. We believe that both theoretical and empirical follow-up work are crucial in AI alignment, with active learning being a particularly important area of focus.

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

# APPENDIX

## A    PROOF OF THEOREM 4

*Proof.* Since $\mathcal{P}_\phi$ is $(3, L, R, U, \beta)$-well-behaved over $(\phi(x), \phi(x'))$, we consider a special 3-dimensional subspace $V$ with its basis being $\vec{u_1} = (\vec{w}^*, \vec{w}^*), \vec{u_2} = (\vec{v_2}, \vec{v_2}), \vec{u_3} = (\vec{v_3}, \vec{v_3})$, where $\vec{v_2}, \vec{v_3}$ are two orthonormal vectors lying on the hyperplane defined by our utility parameter $\vec{w}^*$. Since $\vec{w}^{*T}\vec{v_2} = \vec{w}^{*T}\vec{v_3} = \vec{v_2}^T\vec{v_3} = 0$, these three vectors are linearly independent and form a basis.

Recall that for any point $\pi(x) = (\phi(x), \phi(x'))$ viewed as in the product space $\mathcal{X}^2$, its projected coordinates in $V$ is calculated as $\pi(x)_V = (\vec{u_1}^T \pi(x)) \frac{\vec{u_1}}{||\vec{u_1}||} + (\vec{u_2}^T \pi(x)) \frac{\vec{u_2}}{||\vec{u_2}||} + (\vec{u_3}^T \pi(x)) \frac{\vec{u_3}}{||\vec{u_3}||} = (\vec{u_1}^T \pi(x), \vec{u_2}^T \pi(x), \vec{u_3}^T \pi(x))$.

First, by the subexponential concentration property, for any fixed $h > 0$, $\Pr_{\pi(x) \sim \mathcal{P}_\phi} \left( |\vec{u_2}^T \pi(x)| \geq h \right) < \exp(1 - h/\beta)$, and $\Pr_{\pi(x) \sim \mathcal{P}_\phi} \left( |\vec{u_3}^T \pi(x)| \geq h \right) < \exp(1 - h/\beta)$. So with probability at most $2 \exp(1 - h/\beta)$, the second or the third coordinate of the projection is not within $[-h, h]$.

Next, by the anti-concentration property, the p.d.f. of the projection $\pi(x)_V$ is bounded by a constant $\gamma_V(\pi(x)) \leq U$. Hence the total probability of the projection $\pi(x)_V$'s first coordinate between $[-h, h]$ is $Pr_{\pi(x) \sim \mathcal{P}_\phi} \left( |\vec{u_1}^T \Delta_\phi(x, x')| \leq h \right) \leq \int_{-h}^{h} \int_{-h}^{h} \int_{-h}^{h} U \, dt_1 dt_2 dt_3 + 2 \exp(1 - h/\beta) = 8Uh^3 + \exp(1 - h/\beta)$. Since for any $\phi(x) < 1$, $\exp(\phi(x)) < 1 + \phi(x) + \phi(x)^2$, and $1 - h/\beta < 1$, $\exp(1 - h/\beta) < 1 + ((1 - h/\beta) + (1 - h/\beta)^2)$. So $Pr_{\pi(x) \sim \mathcal{P}_\phi} \left( |\vec{u_1}^T \Delta_\phi(x, x')| \leq h \right) \leq p_1(h)$ has a degree-3 polynomial upper bound.

Now, consider a $1 - D$ projection of this subspace through $\mathrm{Proj}(\phi(x), \phi(x')) = \phi(x') - \phi(x)$. We get $\mathrm{Proj}(V) = \{(\vec{w}^{*T} \Delta_\phi(x, x'), \vec{v_2}^T \Delta_\phi(x, x'), \vec{v_3}^T \Delta_\phi(x, x'))\}$. Due to the triangle and Cauchy-Schwartz inequality, $|\vec{w}^{*T} \Delta_\phi(x, x')| = |\vec{w}^{*T} \phi(x') - \vec{w}^{*T} \phi(x)| \leq |\vec{w}^{*T} \phi(x')| + |\vec{w}^{*T} \phi(x)| \leq \sqrt{2} |\vec{u_1}^T \pi(x)|$. Hence, the probability of the first coordinate of $P(V)$, i.e. the margin $\vec{w}^{*T} \Delta_\phi(x, x')$ being between $[-\sqrt{2}h, \sqrt{2}h]$ has a polynomial upper bound:

$$\Pr_{\Delta_\phi(x, x') \sim \mathcal{P}'_\phi} \left( |\vec{w}^{*T} \Delta_\phi(x, x')| \leq \sqrt{2}h \right) \leq \Pr_{\pi(x) \sim \mathcal{P}_\phi} \left( |\vec{u_2}^T \pi(x)| \geq h \right) < p_1(h).$$

Because $\Pr_{\Delta_\phi(x, x') \sim \mathcal{P}'_\phi} \left( \eta(\Delta_\phi(x, x')) \geq \frac{1}{2} - t \right) = \Pr_{\Delta_\phi(x, x') \sim \mathcal{P}'_\phi} \left( |\vec{w}^{*T} \Delta_\phi(x, x')| \leq F^{-1}(\frac{1}{2} - t) \right)$, as long as $F^{-1}(\frac{1}{2} - t)$ has a polynomial upper bound $p_2(t)$, we could establish another polynomial upper bound $\Pr_{\Delta_\phi(x, x') \sim \mathcal{P}'_\phi} \left( |\vec{w}^{*T} \Delta_\phi(x, x')| \leq F^{-1}(\frac{1}{2} - t) \right) < p_1(\frac{F^{-1}(\frac{1}{2} - t)}{\sqrt{2}}) \leq p_1(\frac{p_2(t)}{\sqrt{2}}) \in poly(t)$.

In other words, we could bound $\Pr_{\Delta_\phi(x, x') \sim \mathcal{P}'_\phi} \left( |\vec{w}^{*T} \Delta_\phi(x, x')| \leq F^{-1}(\frac{1}{2} - t) \right) \leq At^{\frac{\alpha}{1-\alpha}}$ by taking the leading coefficient of $p_1(\frac{p_2(t)}{\sqrt{2}})$ being $A$, and $\frac{\alpha}{1-\alpha}$ being the degree of $p_1(p_2(t)) + 1$. Our noise model satisfies the $(\alpha, A)-$Tsybakov noise condition, and the algorithm in Diakonikolas et al. (2021) applies. $\square$

**Proposition 9.** *The inverse of the c.d.f. for the Bradley-Terry model satisfies $F^{-1}(x) \leq \mathrm{poly}(x)$ on $(0, \frac{1}{2}]$.*

*Proof.* The inverse of the standard logistic function $F(x) = \frac{1}{1 + \exp(-x)}$ is the logit function $F^{-1}(x) = logit(x) = \ln(\frac{x}{1-x})$. Because the derivative of the logit function $\frac{1}{x - x^2}$ is monotonically decreasing from $\infty$ to $4$ on $(0, \frac{1}{2}]$, the logit function is concave. Hence, it is bounded above by its gradient at $x = \frac{1}{2}$, which is $4x - 2$. We have found a polynomial upper bound for $F^{-1}(x) \leq 4x - 2$ for $x \in (0, \frac{1}{2}]$. $\square$

**Proposition 10.** *The inverse of the c.d.f. for the Thurstone-Mosteller model satisfies $F^{-1}(x) \leq \mathrm{poly}(x)$ on $(0, \frac{1}{2}]$.*

*Proof.* The inverse of the standard Gaussian c.d.f function $F(x) = \frac{1}{2}(1 + \mathrm{erf}(\frac{x}{\sqrt{2}}))$ is $F^{-1}(x) = \sqrt{2}\mathrm{erf}^{-1}(2x - 1)$, where erf is the error function. As the derivative of this function is $\sqrt{2\pi} \exp(\mathrm{erf}^{-1}(2x -$

$1)^2$) , which is monotonically decreasing on $(0, \frac{1}{2}]$, the function is concave and bounded above by the gradient at $x = \frac{1}{2}$, which is $\sqrt{2\pi}(x - \frac{1}{2})$. Hence $F^{-1}(x) = \sqrt{2}\mathrm{erf}^{-1}(2x - 1) \leq \sqrt{2\pi}(x - \frac{1}{2})$ for $x \in (0, \frac{1}{2}]$. $\quad\square$

## B    PROOF OF THEOREM 5

*Proof.* Recall that $\Pr(x' \succ x) = F(\vec{w}^{*T}\Delta_\phi(x, x')) = 1 - F(-\vec{w}^{*T}\Delta_\phi(x, x')) = 1 - \Pr(x \succ x')$. First, we show the strong convexity of the loss function

$$
\ell(\vec{w}) = -\frac{1}{n}\sum_{i=1}^{n} \log\left(1(y^i = 1) \cdot \Pr(x' \succ x) + 1(y^i = 0) \cdot \Pr(x' \prec x)\right)
$$

$$
= -\frac{1}{n}\sum_{i=1}^{n} \log\left(1(y^i = 1) \cdot F(\vec{w}^T\Delta_\phi(x, x')) + 1(y^i = 0) \cdot F(-\vec{w}^T\Delta_\phi(x, x'))\right).
$$

Its gradient and Hessian are

$$
\nabla\ell(\vec{w}) = -\frac{1}{n}\left[\sum_{i=1}^{n}\left(1(y^i = 1) \cdot \frac{F'(\vec{w}^T\Delta_\phi(x, x'))}{F(\vec{w}^T\Delta_\phi(x, x'))} + 1(y^i = 0) \cdot \frac{F'(-\vec{w}^T\Delta_\phi(x, x'))}{F(-\vec{w}^T\Delta_\phi(x, x'))}\right)\right]\Delta_\phi(x, x')
$$

$$
\nabla^2\ell(\vec{w}) = \frac{1}{n}\sum_{i=1}^{n}(1(y^i = 1) \cdot \frac{F'(\vec{w}^T\Delta_\phi(x, x'))^2 - F''(\vec{w}^T\Delta_\phi(x, x')) \cdot F(\vec{w}^T\Delta_\phi(x, x'))}{F(\vec{w}^T\Delta_\phi(x, x'))^2}
$$

$$
+ 1(y^i = 0) \cdot \frac{F'(-\vec{w}^T\Delta_\phi(x, x'))^2 - F''(-\vec{w}^T\Delta_\phi(x, x')) \cdot F(-\vec{w}^T\Delta_\phi(x, x'))}{F(-\vec{w}^T\Delta_\phi(x, x'))^2}) \cdot \Delta_\phi(x, x')\Delta_\phi(x, x')^T
$$

By assumption, we have $F'(z)^2 - F''(z) \cdot F(z) \geq \gamma > 0$. Then we can derive strong convexity of $\ell$:

$$
v^T\nabla^2\ell(w)v \geq \frac{\gamma}{n}||Xv||_2^2 \quad \text{for all } v,
$$

where $X$ has $\Delta_\phi(x_i)$ as its $i$-th row, yielding

$$
\ell(\hat{\vec{w}}) - \ell(\vec{w}^*) - \langle\nabla\ell(\vec{w}^*), \hat{\vec{w}} - \vec{w}\rangle \geq \frac{\gamma}{n}||X(\hat{\vec{w}} - \vec{w}^*)||_2^2 = \gamma||\hat{\vec{w}} - \vec{w}^*||_\Sigma^2.
$$

Since $\hat{\vec{w}}$ is the optimal for $\ell$,

$$
\ell(\hat{\vec{w}}) - \ell(\vec{w}^*) - \langle\nabla\ell(\vec{w}^*), \hat{\vec{w}} - \vec{w}^*\rangle \leq -\langle\nabla\ell(\vec{w}^*), \hat{\vec{w}} - \vec{w}^*\rangle.
$$

Then as

$$
|\langle\nabla\ell(\vec{w}^*), \hat{\vec{w}} - \vec{w}^*\rangle| \leq ||\nabla\ell(\vec{w}^*)||_{(\Sigma + \lambda I)^{-1}}||\hat{\vec{w}} - \vec{w}^*||_{\Sigma + \lambda I},
$$

we now would like to bound the term $||\nabla\ell(\vec{w}^*)||_{(\Sigma + \lambda I)^{-1}}$.

Note the gradient $\nabla\ell(w^*)$ can be viewed as a random vector $V \in \mathbb{R}^n$ with independent component:

$$
V_i = \begin{cases} \frac{F'(\vec{w}^{*T}\Delta_\phi(x_i))}{F(\vec{w}^{*T}\Delta_\phi(x_i))} & \text{w.p. } F(\vec{w}^{*T}\Delta_\phi(x_i)) \\ \frac{F'(-\vec{w}^{*T}\Delta_\phi(x, x'))}{F(-\vec{w}^{*T}\Delta_\phi(x_i))} & \text{w.p. } F(-\vec{w}^{*T}\Delta_\phi(x_i)) \end{cases}.
$$

We know that $F(\vec{w}^T \Delta_\phi(x_i)) + F(-\vec{w}^T \Delta_\phi(x_i)) = \Pr(x' \succ x) + \Pr(x \succ x') = 1$, taking the derivative of this equation, we can conclude that the expected value is zero $\mathbb{E}[V] = F'(\vec{w}^T \Delta_\phi(x_i)) + F'(-\vec{w}^T \Delta_\phi(x, x')) = 0$.

And because $F$ is the c.d.f., and $F'$ is the p.d.f, by definition $F'(z) < F(z)$, and $\frac{F'(\vec{w}^T \Delta_\phi(x_i))}{F(\vec{w}^T \Delta_\phi(x_i))} < 1$, $\frac{F'(-\vec{w}^T \Delta_\phi(x, x'))}{F(-\vec{w}^T \Delta_\phi(x_i))} < 1$.

Hence all the variables $V_i$ are 1-sub-Gaussian, and the Bernstein's inequality in quadratic form applies (see e.g. Hsu et al. (2012) (Theorem 2.1)) implies that with probability at least $1 - \delta$,

$$||\nabla \ell(\vec{w}^*)||^2_{(\Sigma + \lambda I)^{-1}} = V^T M V \leq C_1 \cdot \frac{d + \log(1/\delta)}{n},$$

where $C_1$ is some universal constant and $M = \frac{1}{n^2} X (\Sigma + \lambda I)^{-1} X^T$.

Furthermore, let the eigenvalue decomposition of $X^T X$ be $U \Lambda U^T$. Then we can bound the trace and operator norm of $M$ as

$$Tr(M) = \frac{1}{n^2} Tr(U(\Lambda/n + \lambda I)^{-1} U^T U \Lambda U^T) \leq \frac{d}{n}$$

$$Tr(M^2) = \frac{1}{n^4} Tr(U(\Lambda/n + \lambda I)^{-1} U^T U \Lambda U^T) U (\Lambda/n + \lambda I)^{-1} U^T U \Lambda U^T) \leq \frac{d}{n^2}$$

$$||M||_{op} = \lambda_{max}(M) \leq \frac{1}{n}.$$

This gives us,

$$\gamma ||\hat{\vec{w}} - \vec{w}^*||^2_\Sigma \leq ||\nabla \ell(\vec{w}^*)||_{(\Sigma + \lambda I)^{-1}} ||\hat{\vec{w}} - \vec{w}^*||^2_\Sigma$$
$$\leq \sqrt{C_1 \cdot \frac{d + \log(1/\delta)}{n}} ||\hat{\vec{w}} - \vec{w}^*||^2_\Sigma.$$

Solving the above inequality gives us,

$$||\hat{\vec{w}} - \vec{w}^*||_\Sigma \leq C_2 \cdot \sqrt{\frac{d + \log(1/\delta)}{n}}.$$

$\square$

## C  ALGORITHM 1

### C.1  PROOF OF THEOREM 7

*Proof.* Consider Algorithm 1: it runs binary search on each of the $m - 1$ dimensions. The goal is to shrink the version space into a $m - 1$ dimensional hypercube with side length $2\varepsilon/\sqrt{m-1}$. As the longest possible distance between its centre and any point in the cube is the half diagonal $\frac{\sqrt{m-1}}{2} \cdot \frac{2\varepsilon}{\sqrt{m-1}} = \varepsilon$, returning $\hat{\vec{w}}$ as the centre of the cube will suffice the bounded error $e_2 \leq \varepsilon$.

---

**Algorithm 1** Noise-Free Active Learning

---

**Input**: dimension of the instance space $m$, error bound $\varepsilon$
**Output**: $\hat{u}$ with $e_2(\hat{u}, u) \leq \varepsilon$

1: Initialize the Cartesian coordinate system $\mathcal{C}_1$ for $\mathbb{R}^m$
2: Initialise the version space $\mathcal{W} = \{\vec{w} \in \mathbb{R}^m \mid \vec{w} \geq 0, \|\vec{w}\|_1 = 1\}$
3: Initialize a separate $m - 1$ dimensional Cartesian coordinate system $\mathcal{C}_2$ for $\mathcal{W}$
4: $hypercube \leftarrow []$
5: **for** $i = 1, \ldots, m - 1$ **do**
6:     Let $s$ be the length of the version space $\mathcal{W}$ along the $i$-th axis of $\mathcal{C}_2$
7:     **while** $s > 2\varepsilon/\sqrt{m-1}$ **do**
8:         Let $\vec{h_0}, \vec{h_2}$ be two $m - 2$-dimensional hyperplanes tangent to the top and bottom of $\mathcal{W}$ along the $i$-th axis of $\mathcal{C}_2$
9:         Let $\vec{h_1}$ be the $m - 2$-dimensional hyperplane cutting through the middle of $\mathcal{W}$ along the $i$-th axis of $\mathcal{C}_2$
10:       Let $\vec{v}$ be the outward pointing normal vector of the $m - 1$-dimensional hyperplane in $\mathbb{R}^m$ consisting of $\vec{h_1}$ and the origin of $\mathcal{C}_1$
11:       Define $x = 0$, and compute $x' = \tilde{\phi}^{-1}(\phi(0) + v)$
12:       Let $\vec{h}$ be the $m - 2$ hyperplane in $\mathbb{R}^m$ as the intersection between $\mathcal{W}$ and the $m - 1$ hyperplane normal to $\phi(x') - \phi(x)$
13:       Ask the oracle about $(x, x')$
14:       **if** the label $y = 1$ **then**
15:          $bounds \leftarrow \{\vec{h_0}, \vec{h}\}$
16:          $\mathcal{W} \leftarrow$ the half of $\mathcal{W}$ between $\vec{h_0}$ and $\vec{h}$
17:       **else**
18:          $bounds \leftarrow \{\vec{h}, \vec{h_2}\}$
19:          $\mathcal{W} \leftarrow$ the half of $\mathcal{W}$ between $\vec{h}$ and $\vec{h_2}$
20:       **end if**
21:       $s \leftarrow s/2$
22:     **end while**
23:     $hypercube$.append($bounds$)
24: **end for**
25: **return** the center of $hypercube$

---

In order to shrink the length of the version space in each dimension, we recursively select the query $(x, x')$ with difference $\Delta_\phi(x, x')$ being the normal vector of the hyperplane (approximately) halving the original space along that dimension. Let us call the halving hyperplane $\vec{v}$. Now we generate a pair with $\Delta_\phi(x, x') = \phi(\tilde{\phi}^{-1}(\phi(0) + \vec{v})) - \phi(0)$. Suppose the width of the version space $\mathcal{W}$ along the current axis is $s$. By our presumption, we can bound the error $\|\phi(\tilde{\phi}^{-1}(\phi(0) + \vec{v})) - (\phi(0) + \vec{v})\|_\infty = \|\Delta_\phi(x, x') - \vec{v}\|_\infty < \frac{s}{10}$.

Depending on the noise-free signal, we can cut out at least $\frac{1}{2} - \frac{1}{10} = \frac{2}{5}$ of the space. Since the original version space has its length bounded by 1, in order to reduce it to $2\varepsilon/\sqrt{m-1}$ through binary search, we need to repeat $\mathcal{O}(\log(\frac{\sqrt{m-1}}{\varepsilon}))$ times for each dimension. And there will be $\mathcal{O}(\log(\frac{\sqrt{m-1}}{\varepsilon})(m-1))$ queries in total. $\qquad\square$

# D    ALGORITHM 2

## D.1    PSEUDOCODE FOR ALGORITHM 2

---

**Algorithm 2** Active Learning with Noise

---

**Input**: dimension of the instance space $m$, error bound $\varepsilon$
**Output**: $\hat{u}$ with $e_2(\hat{u}, u) \leq \varepsilon$

1: Initialize the Cartesian coordinate system $\mathcal{C}_1$ for $\mathbb{R}^m$
2: Initialise the version space $\mathcal{W} = \{w \in \mathbb{R}^m \mid w > 0, \|w\|_1 = 1\}$
3: Initialize a separate $m - 1$ dimensional Cartesian coordinate system $\mathcal{C}_2$ for $\mathcal{W}$
4: $hyperplanes \leftarrow []$
5: $p_0 \leftarrow F(\varepsilon/\sqrt{m-1})$
6: **for** $i = 1, \ldots, m - 1$ **do**
7:    Let $d$ be the length of the current version space $\mathcal{W}$ along the $i$-th axis of $\mathcal{C}_2$
8:    **while** $d > 2\varepsilon/\sqrt{m-1}$ **do**
9:       Let $h_0, h_2$ be two $m - 2$-dimensional hyperplanes tangential to the top and bottom of $\mathcal{W}$ along the $i$-th axis of $\mathcal{C}_2$
10:       Let $h_1$ be the $m - 2$-dimensional hyperplane cutting through the middle of $\mathcal{W}$ along the $i$-th axis of $\mathcal{C}_2$
11:       Let $v$ be the outward pointing normal vector of the $m-1$-dimensional hyperplane in $\mathbb{R}^m$ consisting of $h_1$ and the origin of $\mathcal{C}_1$
12:       Let $x = 0$, $x' = \phi^{-1}(\phi(0) + v)$, and $h$ be the $m - 2$ hyperplane in $\mathbb{R}^m$ as the intersection between $\mathcal{W}$ and the $m - 1$ hyperplane normal to $\phi(x') - \phi(x)$
13:       **for** $j = 1, \ldots, T$ **do**
14:          Ask the oracle about $(x, x')$
15:          Update $S_T$ accordingly
16:       **end for**
17:       **if** $|S_T - T/2| > T(p_0 - 1/2)/2$ and $S_T > T/2$ **then**
18:          $bounds \leftarrow \{h_0, h\}$
19:          $\mathcal{W} \leftarrow$ the half of $\mathcal{W}$ between $h_0$ and $h$
20:          $d \leftarrow d/2$
21:       **else if** $|S_T - T/2| > T(p_0 - 1/2)/2$ and $S_T < T/2$ **then**
22:          $bounds \leftarrow \{h, h_2\}$
23:          $\mathcal{W} \leftarrow$ the half of $\mathcal{W}$ between $h$ and $h_2$
24:          $d \leftarrow d/2$
25:       **else**
26:          $hyperplane \leftarrow h$
27:          **break**
28:       **end if**
29:    **end while**
30:    **if** $hyperplane$ is undefined **then**
31:       $hyperplanes$.add(the hyperplane with equal distance to the bounds)
32:    **else**
33:       $hyperplanes$.add($hyperplane$)
34:    **end if**
35: **end for**
36: **return**  an intersection point of the whole $hyperplanes$

---

## D.2 Proof of Theorem 8

*Proof.* The high-level procedure of Algorithm 2 is as follows: for every query we constructed in Algorithm 1, we repeat the query $T$ times. Let us denote the sum of a query's labels after $T$ repetitions by $S_T$. If the majority of the labels is ambiguous, i.e., $|S_T - \frac{T}{2}| \leq \frac{T(p_0 - 1/2)}{2}$ then we know $\vec{w}$ is close enough to the hyperplane determined by our query with high probability. Otherwise, we opt to trust the majority vote and halve the search space until the distance between our two hyperplanes is smaller than our required length. We finish after looping through every dimension.

Now we start calculating the upper bound of times we need to ask every turn. Without loss of generality, we assume that the true label of the query is 1. Then the expected value $\mathbb{E}(S_T) > T/2$. If we have made any query cutting through the final hypercube that we want to construct with side length $\frac{2\varepsilon}{\sqrt{m-1}}$, the highest chance of getting a true label will be $p_0 = F(\frac{\varepsilon}{\sqrt{m-1}}) > 1/2$ since its margin with respect to $\vec{w^*}$ could only be even smaller than $\frac{\varepsilon}{\sqrt{m-1}}$.

First, we would like to verify that if we have witnessed $|S_T - \frac{T}{2}| \leq \frac{T(p_0 - 1/2)}{2}$, with high probability we have already queried a hyperplane within a small margin to $\vec{w^*}$. By Hoeffding's inequality, we have $\Pr\left(|\mathbb{E}(S_T) - S_T| \geq \frac{T(p_0 - 1/2)}{2}\right) \leq \exp(-\frac{T(p_0 - 1/2)^2}{4})$. So with probability at least $1 - \exp(-\frac{T(p_0 - 1/2)^2}{4})$, $|\mathbb{E}(S_T) - S_T| \leq \frac{T(p_0 - 1/2)}{2}$. Then the triangle inequality gives us $|\mathbb{E}(S_T) - \frac{T}{2}| = |\mathbb{E}(S_T) - S_T + S_T - \frac{T}{2}| \leq |\mathbb{E}(S_T) - S_T| + |S_T - \frac{T}{2}| \leq \frac{T(p_0 - 1/2)}{2} + \frac{T(p_0 - 1/2)}{2} = T(p_0 - \frac{1}{2})$ with probability at least $1 - \exp(-\frac{T(p_0 - 1/2)^2}{4})$. In other words, with high probability, $\frac{T}{2} < \mathbb{E}(S_T) \leq T(p_0)$. Since $\mathbb{E}(S_T)$ is determined by the distance between the query and $\vec{w^*}$, we deduce that the current query satisfies a small margin as wanted, hence we can stop.

Next, we would like to confirm the majority is the true label with high confidence if we have witnessed a majority vote with significance, i.e., when $|S_T - \frac{T}{2}| > \frac{T(p_0 - 1/2)}{2}$. Recall again with our true label being 1, $\mathbb{E}(S_T) > \frac{T}{2}$. The probability of witnessing a false majority is $\Pr\left(S_T - \frac{T}{2} < -\frac{T(p_0 - 1/2)}{2}\right) = \Pr\left(\frac{T}{2} - S_T > \frac{T(p_0 - 1/2)}{2}\right) \leq \Pr\left(\mathbb{E}(S_T) - S_T \geq \frac{T(p_0 - 1/2)}{2}\right) \leq \exp(-\frac{T(p_0 - 1/2)^2}{4})$ since $\mathbb{E}(S_T) - S_T > \frac{T}{2} - S_T$.

From Theorem 7, we know we will need $\mathcal{O}(\log(\frac{\sqrt{m-1}}{2\varepsilon})(m-1))$ accurate query labels through votes. So the cumulative confidence is to satisfy

$$q^{\mathcal{O}(\log(\frac{\sqrt{m-1}}{2\varepsilon})(m-1))} \geq 1 - \delta,$$

where $q$ is the success rate for each query on a different hyperplane. Therefore $q \geq (1-\delta)^{\frac{1}{\mathcal{O}(\log(\frac{\sqrt{m-1}}{2\varepsilon})(m-1))}}$. And for each turn we just need $\exp(-\frac{T(p_0 - 1/2)^2}{4}) \leq 1 - q$. By a simple calculation, we get the sufficient number of repetitions from $T \geq T_0 = -\frac{4}{(p_0 - 1/2)^2} \log(1 - (1-\delta)^{\frac{1}{\mathcal{O}(\log(\frac{\sqrt{m-1}}{2\varepsilon})(m-1))}})$.

Our final sample complexity is $\mathcal{O}(T_0 m \log(\frac{\sqrt{m-1}}{2\varepsilon})) = \mathcal{O}((\frac{1}{F(\frac{\varepsilon}{\sqrt{m-1}}) - 1/2)^2} \log(\frac{1}{\delta})))$. □

## D.3 Sample complexity for active learning of Bradley-Terry Model

**Corollary 1.** *For the standard logistic function $F(x) = \frac{1}{e^{-x}+1}$, the sample complexity of learning $e_2$ is polynomial in $(\frac{1}{\varepsilon}, \log(\frac{1}{\delta}), m)$.*

*Proof.* The sample complexity is $\mathcal{O}(\frac{1}{(F(\frac{\varepsilon}{\sqrt{m-1}})-\frac{1}{2})^2 \log(\frac{1}{\delta})}) = \mathcal{O}(\frac{1}{F(\frac{\varepsilon}{\sqrt{m-1}})^2 \log(\frac{1}{\delta})}) = \mathcal{O}((1 + \exp(-\frac{\varepsilon}{\sqrt{m}})^2 \log(\frac{1}{\delta})) = \mathcal{O}(\exp(\frac{-\varepsilon}{\sqrt{m-1}})^2 \log(\frac{1}{\delta}))$. Observe that for any $m \geq 2$ and $0 < \varepsilon < 1$, $\frac{\varepsilon}{\sqrt{m-1}} > 0$, yielding $\varepsilon < 1 < \exp(\frac{\varepsilon}{\sqrt{m-1}}) < \exp(\frac{\varepsilon}{\sqrt{m-1}})^2$, implying $\frac{1}{\exp(\frac{\varepsilon}{\sqrt{m-1}})^2} \log(\frac{1}{\delta}) < \frac{1}{\varepsilon} \log(\frac{1}{\delta})$. Therefore, the sample complexity for logistic noise is in $\mathcal{O}(\frac{1}{\varepsilon} \log(\frac{1}{\delta}))$. $\square$

Ccomparing this result with the passive learning result from inequality (3), whose sample complexity is $\mathcal{O}(\frac{1}{\varepsilon} \log(\frac{1}{\delta}) + \frac{m}{\varepsilon})$, our sample complexity bound is evidently better because it doesn't have the second term $\frac{m}{\varepsilon}$.

## D.4 NUMERICAL ILLUSTRATION

In order to illustrate the sample complexity gap between passive and active learning, we conducted the following experiment. First, we assume that the pairwise preference labels are generated according to the Bradley-Terry model. For passive learning, we apply the robust logistic regressor directly on the dataset with the pairs' difference $\Delta_\phi(x, x')$ scatter around the orthogonals of the ground truth weight $\vec{w}^*$; for active learning, we implemented and ran our active learning algorithm 2 directly.

The plot demonstrates clearly that active learning could achieve better accuracy and higher confidence with much fewer samples.

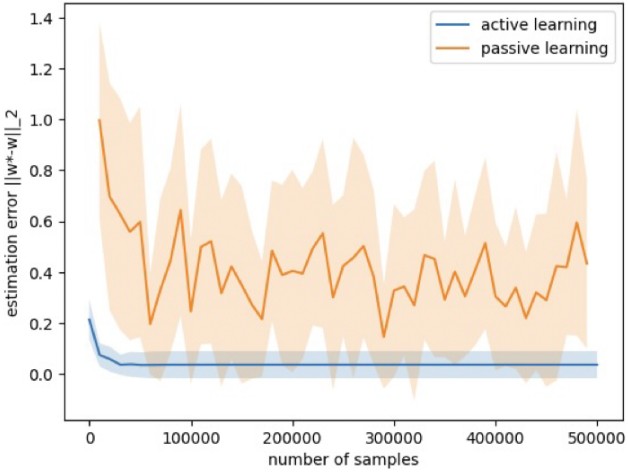

Figure 2: Sample Complexity Comparison Between Passive and Active Learning.

