# OpenReview forum: "Learning Linear Utility Functions From Pairwise Comparison Queries"
_ICLR.cc/2025/Conference — Submitted to ICLR 2025_

### Official Review · Reviewer_92sg · 2024-10-27

**Soundness:** 3
**Presentation:** 3
**Contribution:** 3
**Rating:** 6
**Confidence:** 3

**Summary:**

The paper explores the feasibility of learning linear utility functions from pairwise comparison queries, proposing that this method can be applied to improve AI alignment with human values. The study addresses two levels of the problem: predicting out-of-sample pairwise comparisons and accurately estimating utility parameters. In the passive learning setting, it demonstrates that predicting comparisons is feasible, but estimating utility parameters is challenging without strong structural noise assumptions. The negative result, indicating that the linear utility function is not PAC-PC learnable, is presented in this setting. In the active learning setting, where comparisons can be strategically selected, estimating utility parameters becomes much more feasible, provided there is the ability to invert the embedding.

**Strengths:**

Please find the strengths below:
1. The problem of learning the linear utility function is important in AI. The paper addresses both passive and active learning settings, which are commonly encountered in practice.
2. For passive learning, the paper presents a negative result regarding the learnability of linear utility function parameters.
3. The proofs in the main paper are clear and intuitive to follow.

**Weaknesses:**

Please find the weaknesses below:
1. The paper lacks intuitive explanations for these assumptions, such as examples of cases where the assumptions are satisfied.
2. The paper does not sufficiently discuss the practicality of assumptions, such as those on well-behaved isotropic distributions, $F'(z)^2 - F''(z) \cdot F(z) \geq \gamma$, and the approximation of the inverse of $\phi$. It is unclear how to determine parameters (e.g., $c$ and $\gamma$) in these assumptions in practice.
3. In Theorems 3, 7, and 8, the paper only establishes the existence of an efficient algorithm but does not explain how to find the algorithm within the main text (though pseudo code for Theorems 7 and 8 is provided in the appendix).
4. The paper lacks experiments assessing the complexity of the algorithms. Even some numerical experiments would be beneficial, as would verification of these assumptions on practical datasets.

**Questions:**

Refer to the weaknesses section above and find the additional questions below:
1. Is there a lower bound for the sample complexity in Theorem 3? It seems that the complexity could be exponentially high with respect to $1/\alpha$.
2. Is it possible to extend beyond linear utility functions, particularly for the simpler problem of predicting pairwise comparisons?

**Details Of Ethics Concerns:**

I did not identify any ethical concerns.

---

> ### Author Response · Authors · 2024-11-17
> **Thank you for your thoughtful and detailed comments and questions, which we address below.**
>
> Thank you for your thoughtful and detailed comments and questions, which we address below.
>
>
>
> >**Comment**: The paper lacks experiments assessing the complexity of the algorithms. Even some numerical experiments would be beneficial, as would verification of these assumptions on practical datasets.
>
> **Response:** Since the purpose of this paper was to set up a framework for systematically understanding alignment in terms of learnability formally, we focused on  providing a comprehensive theoretical landscape. However, to address this concern, we added a numerical experiment in Section D4 of the Appendix which illustrates the significant practical value of active learning in enabling qualitatively better learning performance (in particular, we observe that while standard learning approaches fail to effectively learn the reward model parameters even as the amount of data increases, our active learning algorithm successfully learns these).
>
>
> > **Comment**: The paper does not sufficiently discuss the practicality of assumptions...
>
> **Response:**
>
>
> - **Well-behaved distribution**: The assumptions means, essentially, that the data are: 1) overly concentrated (if the mass is all on one extremely similar pair, the data would be very uninformative), and 2) not too spread out (so that we can actually derive a good understanding from the observations).
>
>
>
> - **F’(z)^2 —F(z)F”(z) > 0**: Indeed this is a very strong requirement in practice, but one which is needed for learning to be possible.  This is precisely what motivates our consideration of active learning, which requires far weaker assumptions to enable effective learning.
>
>
> - **Computing the inverse of the embedding**: Like other papers considering learning issues in value alignment, our focus is on sample complexity. Since we assume that $\phi$ is known, computing its inverse amounts does not affect sample complexity, although it is, of course, an important consideration in practice.
>
>
> We have also updated the paper to clarify these points.
>
> >**Comment**: In Theorems 3, 7, and 8, the paper only establishes the existence of an efficient algorithm but does not explain how to find the algorithm within the main text.
>
>
> **Response:** For Theorem 3, note that we are simply referencing this result from another paper, and use it as a building block for Theorem 4 (whose main purpose is to prove that the original algorithm works in our setting as well). For Theorems 7 and 8, we had to move the algorithms to the Appendix due to the space constraint. As our main focus is on sample complexity, we had the main results highlighted in the main body, and computational details were available for those who wish to delve deeper. Nevertheless, we succeeded in finding space and added more descriptions of the algorithms into the revised version.
>
> >**Comment**: Is there a lower bound for the sample complexity in Theorem 3? It seems that the complexity could be exponentially high with respect to $(1/\alpha)$.
>
> **Response**: In the passive learning case, the lower bound is indeed the noiseless case. Note that $\alpha$ is the parameter for the noise model:$\Pr_{x \sim D_x} [\eta(x) \geq \frac{1}{2} - t] \leq A t^{\frac{\alpha}{1-\alpha}}$ If we want the noise to decrease quickly, then $\frac{\alpha}{1-\alpha}$ should be large, or alternatively, $\alpha$ should be closer to 1 rather than 0.
> Thus, in the benign case, we have $O(\frac{1}{\alpha}) = O(1)$, and the sample complexity in Theorem 3 becomes:
>
> $$\beta^4 \left( \frac{dUA}{rl\epsilon} \right)^{O\left(\frac{1}{\alpha}\right)} \log \left( \frac{1}{\delta} \right) = \beta^4 \left( \frac{dUA}{rl\epsilon} \right)^{o(1)} \log \left( \frac{1}{\delta} \right),$$
> which matches the bound we showed in Theorem 1, which is of degree one in terms of $\frac{1}{\epsilon}$ and $\log(1/\delta)$.
> Indeed, if the noise is extremely high, no algorithm can learn the concept efficiently. However, this shouldn't be a major concern. Combining the results from the proof of Theorem 4, Appendix 6 (lines 611 and 619), and our two illustrating examples below, we find that the degree bound you are concerned with,i.e. $O(\frac{1}{\alpha})$ is typically quite low. To make this more concrete, the degree of Tsybakov noise bound $\frac{\alpha}{1-\alpha}$ is both 3*1+1= 4 for Bradley-Terry and Thustone-Mosteller model. By solving $\frac{\alpha}{1-\alpha}$  we get $\alpha = 0.8$, and the term you are concerned about is $\frac{1}{\alpha} = 1.25$ only.
>
> >**Comment**: Is it possible to extend beyond linear utility functions, particularly for the simpler problem of predicting pairwise comparisons?
>
> **Response:** This is a great question! Note that we have generalized our model to some degree by allowing an arbitrary embedding. A further generalization appears non-trivial, and is a fantastic consideration for follow-up work. For estimation, however, we expect the problem to be much more difficult (indeed, identifiability becomes a major issue, let alone sample complexity).

---

### Official Review · Reviewer_NAHi · 2024-11-01

**Soundness:** 4
**Presentation:** 4
**Contribution:** 2
**Rating:** 6
**Confidence:** 4

**Summary:**

Driven by applications in LLM alignment, this paper studies the problem of learning linear utility functions from pairwise comparisons. The authors provide theoretical results that suggest the following.

- Efficiently PAC-learning utilities that predict pairwise comparison outcomes is possible, except in degenerate cases.
- Learning the utility parameters can fail when the RUM does not satisfy certain conditions. The noiseless case is an example of failure case.

The authors then consider the active learning setting, where queries can be chosen adaptively based on previously observed outcomes. In that setting, it is possible to efficiently PAC-learn utility parameters in the noiseless case.

**Strengths:**

- The paper is particularly well written and easy to follow. The organization is clear, the progression makes sense, and the decisions on what to include in the main text and what to defer to the supplementary material are well thought-through.
- The results appear to be rigorous. I did not check every single proof, but the ones I checked appear to be correct and, in general, the arguments make sense.
- Research on this topic is timely, given the increasing interest in practical applications, and highly relevant to the ICLR community.

**Weaknesses:**

- While the exact problem studied in the paper does not have a lot of prior work (to my knowledge), there is an extensive literature on learning RUMs from pairwise comparisons over a fixed set of items with an independent utility for each item in the set. This can be recast as sparse "one-hot" features in the model studied in the paper.
    - some of the insights the paper provides are well-known in that literature. For example, sample complexity results for the Bradley-Terry model typically depend on the noise via the dynamic range, i.e., the maximum ratio between item utilities. The case of unbounded dynamic range has been studied before. Some references: [1], [2], [3].
    - making connections to this literature by comparing & contrasting results would improve the paper significantly.
- In general, I find the connection between the theory developed in this paper and LLM alignnment a bit weak. The paper could be incredibly impactful if there were concrete take-aways that could inform practice. Are there any?
    - l 082: "Our results thus suggest that conventional reward model [...] may yield potentially misleading estimates". Backing this claim up with real data or at least realistic experiments would be helpful.
- The motivation for the active learning section could be improved.
    - The authors say: "first [...] curates a dataset of input pairs, and only then obtains preference data for these". This is the exact _opposite_ of active learning, and indeed the two papers cited (OpenAI & Anthropic) are examples of passive learning.
    - However there is work showing that active learning can improve LLM alignment techniques, see e.g. [4] and [5], so I see an opportunity to make the point much more convincingly.

Minor feedback:

- l 046: "... over a vector space of outcomes" -> what does this mean?
- l 095: "the restriction of our utility function is qeuivalent to ..." -> I could not make sense of this sentence.
- l 154: "with probability $1-\delta$ such that ..." I think this should be flipped: "such that ... with probability $1-\delta$"
- Theorem 2 seems kind of obvious: if one can make outcomes arbitrarily noisy, it can get arbitrarily hard to learn. Is there a deeper insight I am missing here?

References:

1. Generalized Results for the Existence and Consistency of the MLE in the Bradley-Terry-Luce Model <https://proceedings.mlr.press/v162/bong22a/bong22a.pdf>
2. l-infty-Bounds of the MLE in the BTL Model under General Comparison Graphs <https://proceedings.mlr.press/v180/li22g/li22g.pdf>
3. Active Ranking in Practice: General Ranking Functions with Sample Complexity Bounds <https://citeseerx.ist.psu.edu/document?repid=rep1&type=pdf&doi=52f514508329c951ef9b417ce5243cf40db136bd>
4. Active Preference Learning for Large Language Models <https://arxiv.org/pdf/2402.08114>
5. Direct Language Model Alignment from Online AI Feedback <https://arxiv.org/pdf/2402.04792>

**Questions:**

1. What concrete take-aways should a practitioner doing preference-based LLM alignment get from the paper? Which theoretical findings can infrom practice?
2. Much of the theory seems to exploit the fact that learning from pairwise comparisons in linear RUMs is equivalent to linear binary classification. Which insights are unique to the fact that we observe pairwise comparisons?

---

> ### Author Response · Authors · 2024-11-17
> **We thank the reviewer for the thoughtful comments and suggestions!**
>
> We thank the reviewer for the thoughtful comments and suggestions!
>
> **Response:**
>
>
> >**Comment:**  In general, I find the connection between the theory developed in this paper and LLM alignnment a bit weak... What concrete take-aways should a practitioner doing preference-based LLM alignment get from the paper?
>
> **Response:** In common practice of eliciting pairwise comparison preferences pairs of responses to given prompts are chosen somewhat arbitrarily (e.g., responses generated by existing LLMs), and not optimizes for informativeness (and, thus, sample efficiency). This is somewhat akin to the passive learning settings in which we assume that data passively arises, drawn from some fixed distribution. However, we can in principle be far more selective in the prompt responses we present to humans for evaluation (for example, we can use rejection sampling techniques) so as to maximize the informativeness of the responses. Our results show that such active learning approaches can exhibit qualitatively better sample complexity than passive learning, and the current more passive approaches for LLM value alignment may be a significant missed opportunity. We added a brief discussion of this connection in the related work section, as well as (by way of stronger motivation) at the beginning of the section on active learning.
>
> >**Comment:** Which insights are unique to the fact that we observe pairwise comparisons?
>
>
> **Response:**
> While the linearity assumption in our work connects many of our results to the halfspace learning literature, we found our problem to be far from a reduction of it.
>
> In the context of passive learning, as we’ve demonstrated, most Random Utility Models (RUMs) do not imply PAC-PC learnability. Therefore, developing a general yet tractable model that aligns with human psychology presents a unique and complex task.
>
> In the active learning setting, we believe the difference of learning from pairwise comparison data will also emerge in practice. Here, it is critical (due to noise) to be able to ask the same query $(x,x')$ many times. In conventional learning, where a query is just a feature vector $x$, this would be practically challenging, since there is no reason to expect the same person to answer the same question differently each time. In our setting (supposing for illustration that the embedding $\phi$ is an identity), what we actually need to elicit many responses to is $\Delta x = (x'-x)$. Consequently, we can ask an arbitrary number of distinct queries $(x,x')$ that are equivalent from the perspective of learning from pairwise comparisons, but which are not perceived as identical (and would thus elicit a meaningful distribution in practice). The complement of this is that there are also many more ways for data to be uninformative in the pairwise comparison setting, which implies that distributional conditions for both positive and negative results need to be adjusted to this setting for both positive and negative results in the passive learning setting.
> >**Comment:**  While the exact problem studied in the paper does not have a lot of prior work (to my knowledge), there is an extensive literature on learning RUMs from pairwise comparisons over a fixed set of items with an independent utility for each item in the set. This can be recast as sparse "one-hot" features in the model studied in the paper.
>
>
> **Response:** We agree that this is a special case of our setting, but the key distinction between this setting and ours is that we are concerned with generalization beyond the set of feature vectors in the dataset of pairwise comparisons.
>
> >**Comment:** The motivation for the active learning section could be improved.
>
>
> **Response:**  We added a brief discussion of the recent work on active learning in RLHF to bolster the motivation for a more foundational theoretical analysis that we undertake.   Thanks for your correction for active learning's motivation. We have implemented this change in the revised version.
>
> **Minor feedback:**
> > "... over a vector space of outcomes"
>
> We just mean that our setting, in which outcomes ("candidates") correspond to feature vectors, is qualitatively distinct from the typical setting in social choice, in which the set of outcomes (candidates) is explicitly enumerated.
>
> > "the restriction of our utility function..."
>
> Yes, this is very badly worded. We just mean to restrict the parameters to be weight-normalized. This has been fixed in the revised version.
>
> > "with probability ..."
>
> Indeed, this is a typo. It is fixed in the revised version.
>
> > "Theorem 2 seems kind of obvious..."
>
> Our central goal was to offer a comprehensive investigation of the problem of reward model learning that is central to value alignment (which, in turn, has emerged as a major foundational as well as practical concern in modern AI).    Hence, we include natural results for the sake of completeness.

---

### Official Review · Reviewer_v2zX · 2024-11-04

**Soundness:** 3
**Presentation:** 3
**Contribution:** 3
**Rating:** 6
**Confidence:** 3

**Summary:**

This paper studied the learnability of linear utility functions from pairwise comparison queries. The paper considered two settings, including the passive learning setting and the active learning setting, with two objectives: 1) predicting the out-of-sample pairwise preferences; 2) estimating the true parameters of the utility function. Theoretical analysis show that in the passive learning case, the first objective is achievable both when query responses are uncorrupted by noise / under Tsybakov noise when the distributions are sufficiently nice. However the utility function parameters may not be learnable. In contrast, in the active learning setting, both objectives can be efficiently achieved.

**Strengths:**

- This paper studied a significant and practical problem of reward learning, when the observations are pairwise comparison queries. This can be a common scenario in real-world examples such as online recommendation platforms. Overall the paper is presented with a good clarity on the model, assumptions and results.
- The theoretical analysis justified an interesting learnability gap between the passive and active learning settings, where the active queries significantly helped with the utility function estimation. This result may be helpful to provide guidance for real-world applications where the active queries may be obtained via incentivization.

**Weaknesses:**

In the active learning setting, the estimation of the utility function relies on the ability to invert the embedding function, and obtaining the inverse of the embedding can be computationally challenging or infeasible particularly with neural network models.

**Questions:**

- in the active learning setting, is the sample complexity optimal?

---

> ### Author Response · Authors · 2024-11-17
> **We thank the reviewer for the thoughtful comments and suggestions!**
>
> Thank you for your detailed reviews and thoughtful questions!
>
>
> >**Comment**: In the active learning setting, the estimation of the utility function relies on the ability to invert the embedding function, and obtaining the inverse of the embedding can be computationally challenging or infeasible particularly with neural network models.
>
> **Response**: Yes, this is a great point. Since we assume that the embedding is known, the computational complexity of inversion does not impact sample complexity, although it is nevertheless an important open issue.
>
> >**Comment:** In the active learning setting, is the sample complexity optimal?
>
> **Response**: Great question! We are not aware of a corresponding lower bound on the active learning complexity in our setting.

---

> > ### Comment · Reviewer_v2zX · 2024-11-26
> > **reponse**
> >
> > Thank you for your response! I will keep my original score.

---

### Official Review · Reviewer_FKk5 · 2024-11-05

**Soundness:** 3
**Presentation:** 3
**Contribution:** 2
**Rating:** 3
**Confidence:** 4

**Summary:**

This paper addresses PAC learning of linear functions using comparison queries. It considers two objectives: predicting the outcome of a comparison between two inputs and learning the true underlying linear parameters. Both noiseless and noisy settings are explored, as well as both active and passive learning.

**Strengths:**

The problem studied is very clean and interesting to the learning theory community.

The writing is very clear.

**Weaknesses:**

I believe the primary weakness of this paper lies in its technical contributions. I didn't find anything particularly novel or technically interesting.

For instance, the results in passive learning don't appear to be new. Theorems 1 and 3 have been previously addressed, and the observation in Theorem 2 seems straightforward—naturally, certain noisy models prevent learning. Similarly, the conclusion in Theorem 6 also seems obvious.

In the active learning setting, the problem of active learning for halfspaces with comparison queries appears to have already been solved by [1].

[1] Kane, Daniel M., Shachar Lovett, Shay Moran, and Jiapeng Zhang. "Active classification with comparison queries." In 2017 IEEE 58th Annual Symposium on Foundations of Computer Science (FOCS), pp. 355-366. IEEE, 2017.

**Questions:**

It would be nice if the authors could explain the technical novelty of this work.

---

> ### Author Response · Authors · 2024-11-17
> **Our contribution is to unify the framework of computational social choice and RLHF, and to provide a comprehensive view for value alignment**
>
> Thank you for your review and thoughtful questions!  Below, we address all your comments.
>
> > **Comment**: I believe the primary weakness of this paper lies in its technical contributions. I didn't find anything particularly novel or technically interesting. For instance, the results in passive learning don't appear to be new. Theorems 1 and 3 have been previously addressed, and the observation in Theorem 2 seems straightforward—naturally, certain noisy models prevent learning. Similarly, the conclusion in Theorem 6 also seems obvious.
>
> **Response**: Our central goal was to offer _a comprehensive investigation of the problem of reward model learning that is central to value alignment_ (which, in turn, has emerged as a major foundational as well as practical concern in modern AI). Our consideration of passive learning is, thus, to set the context by virtue of the fact that this setting is the starting point in learnability, and was not fully addressed in prior work. For example, recent work in a similar setting has focused solely on the ability to infer the true reward model parameters, whereas we introduce another novel measure---accuracy of pairwise comparison prediction (PAC-PC learnability). Indeed, it is partly this measure of efficacy (error) that enables a natural connection to prior results on learning half spaces.
>
> Unlike previous RLHF work that typically focuses on a single model, our approach involves considering a broad set of Random Utility Models (RUMs). This introduces an added layer of complexity in identifying the most appropriate assumptions for noise distributions in pairwise comparison data. The process of selecting assumptions requires both technical effort and conceptual insight, as finding a right model for pairwise comparisons is inherently challenging.
>
> Our second key conceptual contribution is to contrast the passive learning results with the *active learning* setting: in particular, we observe that our strong positive results for active learning contrast rather dramatically with the rather strong negative results in passive learning.  This is of high practical significance, as value alignment problems often have the ability to carefully select which queries to annotate, but commonly do not fully leverage this ability. Our theoretical results show that this is, potentially, a significant missed opportunity.
>
> On the specifically technical side, we wish to clarify that Theorem 3 is not our contribution (we reference its source), and is provided solely for the paper to be self-contained. Our new result is Theorem 4 (the proof is in the Appendix), and is not straightforward. Theorem 6, in turn, is useful to provide a contrast with the positive result in Theorem 5 (which is significant generalization of the result by Zhu et al). In the active learning setting, the main result is Theorem 8 which provides the key contrast with the general impossibility in the passive learning setting (Theorem 6).  All of our results, thus, provide a reasonably comprehensive picture of learnability of linear reward models from pairwise preference comparisons.
>
> > **Comment**: In the active learning setting, the problem of active learning for halfspaces with comparison queries appears to have already been solved by [1]. [1] Kane, Daniel M., Shachar Lovett, Shay Moran, and Jiapeng Zhang. "Active classification with comparison queries." In 2017 IEEE 58th Annual Symposium on Foundations of Computer Science (FOCS), pp. 355-366. IEEE, 2017.
>
> **Response:** Thank you for pointing out this paper. We note that despite the identical term, our learning model is quite different from Dane et al, so that the results are essentially unrelated. First, Dane et al. are allowed to make *both* label queries (the classification label for a given input) *and* comparison queries (comparing classification scores of two inputs). We can only ask preference comparison queries. Second, we allow query synthesis methods, while their framework is pool-based, meaning queries must be made from a prescribed set. Third, our learning framework explicitly considers noise in preference comparison responses, whereas Dane et al. assume that responses are deterministic.

---

> ### Author Response · Authors · 2024-11-24
> **Discussion period ending soon**
>
> Dear reviewer, we tried to address your concerns in our rebuttal.  As the discussion period is ending in a few days, please let us know if our response addresses these, or if you have additional or residual concerns.

---

> > ### Author Response · Authors · 2024-11-28
> >
> > Dear reviewer, we hope you had a chance to read our responses to your concerns.  Please let us know if you have any additional comments or concerns that our response has not resolved.

---

### Meta-Review · Area_Chair_WPiH · 2024-12-23

**Metareview:**

This paper studies the problems of learning linear utility function from pairwise comparison feedback. This paper gives solid theoretical guarantees in both passive learning and active learning setting, with rigorous sample complexity guarantees. This paper consists a set of solid and interesting theoretical results. However, while the paper motivates the problem from the practical RLHF, how to algorithms and sample complexity derived from this paper are quite unclear. Several algorithms used in this paper are very different from what were used in practice. If authors could also provide a few empirical evidence on the superiority of the proposed methods that can potentially make the work much more impactful. We recommend rejection based on its current form.

**Additional Comments On Reviewer Discussion:**

While this paper gives a set of solid theoretical results, the overall excitement about this paper is low among all reviewers. While motivated from a practical problem, the connection of its results to practice is weak. The technical novelty is not significant enough to overshine other drawbacks.

---

### Decision · Program_Chairs · 2025-01-22

Reject